

# Energy exchange and fluctuations between a dissipative qubit and a monitor under continuous measurement and feedback

**Tsuyoshi Yamamoto⋆ and Yasuhiro Tokura†**

Institute of Pure and Applied Sciences, University of Tsukuba,
Tsukuba, Ibaraki 305-8577, Japan

⋆ yamamoto.tsuyoshi.ts@u.tsukuba.ac.jp , † tokura.yasuhiro.ft@u.tsukuba.ac.jp

## Abstract

Continuous quantum measurement and feedback induce energy exchange between a dissipative qubit and a monitor even in the steady state, as a measurement backaction. Using the Lindblad equation, we identified the maximum and minimum values of the steady-state energy flow as the measurement and feedback states vary, and we demonstrate the qubit cooling induced by these processes. Turning our attention to quantum trajectories under continuous measurement and feedback, we observe that the energy flow fluctuates around the steady-state values. We reveal that the fluctuations are strongly influenced by the measurement backaction, distinguishing them from the standard Poisson noise typically observed in electronic circuits. Our results offer potential application in the development of quantum refrigerators controlled by continuous measurement and feedback, and provide deep insight into quantum thermodynamics from the perspective of fluctuation.

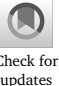

# 1 Introduction

Quantum measurement and feedback induce unique non-unitary dynamics in a measured quantum system [1–3]. This measurement backaction has advanced the fields of quantum thermodynamics [4–6] and the application of quantum thermal machines [7–13]. The heat change in the measured quantum system due to the measurement and the feedback is a key concept in quantum energetics, e.g., work or heat extraction by operating measurement and feedback [14,15]. Recent advances in quantum technology have allowed us to observe the energy exchange between the measured quantum system and the monitor under the measurement, which does not commute with the system Hamiltonian, in superconducting circuits [16]. This experimental progress motivates further theoretical research on the energy exchange by the non-commuting measurements. While the projective quantum measurement and feedback have been extensively studied so far [17–24], recently, the continuous quantum measurement and feedback attract attention from fundamental aspects of quantum thermodynamics, such as the quantum fluctuation theorem [25] and quantum thermodynamics uncertainty relations (QTURs) [26,27], as well as quantum thermal machines [28–30]. Most studies on quantum measurement have been conducted in the context of ideal systems without dissipation. It is crucial to take dissipation into account not only because of its ubiquity in nature but also due to its qualitative effects on energy exchange. This consideration enables us to address steady-state properties of energy flow under continuous quantum measurement and feedback, as measurement and feedback alone induce no energy exchange in the steady-state limit.

Continuous quantum measurement and feedback process is inherently stochastic, as the quantum state probabilistically jumps to a measured state. Consequently, the energy exchange induced by continuous quantum measurement and feedback fluctuates around the ensemble-averaged energy flow with respect to the measurement outcomes. Current fluctuations in quantum transport have been studied in the mesoscopic physics [31,32]. For instance, for non-interacting electrons in a one-dimensional nanowire with weak transmission, the variance of the electronic current, which characterizes current fluctuations, is proportional to the mean electronic current. This is reflected by the fact that free electrons transferring through the nanowire obey the Poisson statistics. Observing current fluctuations provides deeper insights into the microscopic scattering process of carriers. Now, current fluctuations have been studied theoretically and experimentally in strongly correlated systems, such as the fractional quantum Hall effect [33,34], the Kondo effect [35,36], and superconductivity [37–39]. Moreover, since the statistical properties lie behind the current noise, it is closely related to the fluctuation theorem [40,41].

In this paper, we investigate the energy exchange between a monitor and a qubit with dissipation by connecting the qubit to bosonic environments, under continuous quantum measurement and feedback. In the first part, we examine the steady-state energy flow using the Lindblad equation and derive its minimum and maximum values by varying the measurement and feedback states. Reference [42] has demonstrated that, in the absence of feedback, the energy exchange does not occur from the dissipative qubit to the monitor under continuous quantum measurement. In this work, we incorporate a *feedback* sequence following quantum measurement to create and stabilize desired quantum states that cannot be achieved through

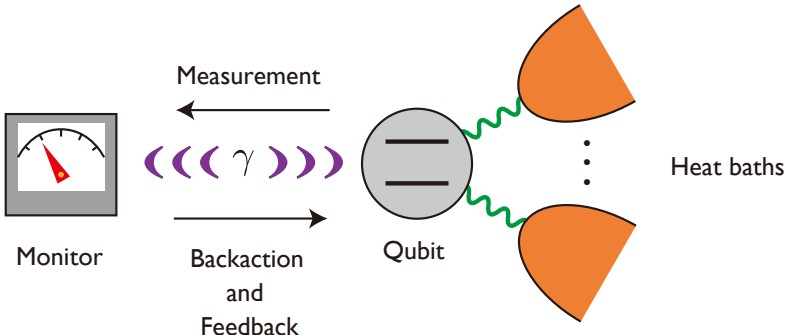

Figure 1: Monitored qubit is coupled to the heat baths. The qubit state is continuously measured with the strength $\gamma$, and affected by the measurement as a backaction and feedback.

measurement alone. As a result, we demonstrate that the monitor can extract energy from the dissipative qubit. This cooling effect could have potential applications in measurement-based quantum refrigerators. The second part focuses on the fluctuations in the energy flow by tracing individual quantum trajectories. We demonstrate that the power spectrum consists of two distinct processes unique to continuous quantum measurement: Poisson noise and backaction noise, associated with quantum jumps and measurement backaction dynamics, respectively. Notably, we observe that the Fano factor, which is the ratio of total noise to Poisson noise, remains below unity across a wide range of parameters. The perspective from fluctuations of the continuous quantum measurement and feedback can lead to the further understating of thermodynamics in quantum dissipative systems.

This paper is organized as follows. In the next section, we introduce the theoretical model used in this study and describe quantum dynamics with both dissipation and continuous measurement and feedback. In Sec. 3, we calculate the steady-state energy flow between the dissipative qubit and the monitor and obtain its minimum and maximum values. In Sec. 4, we address the fluctuations of the energy exchange and analyze the noise power spectra. Finally, the summary is drawn in Sec. 5. Throughout this paper, We set $\hbar = k_B = 1$.

## 2 Model

In this section, we introduce the Hamiltonian of the dissipative qubit and describe the quantum dynamics of the reduced density matrix under continuous quantum measurement and feedback, as shown in Fig. 1.

### 2.1 Dissipative qubit

The dissipation of the qubit is described by the Jaynes-Cummings-type coupling when the qubit is weakly coupled with heat baths. The Hamiltonian is given by

$$H = H_0 + \sum_{rk} \omega_{rk} b_{rk}^\dagger b_{rk} + \sum_{rk} \frac{\lambda_{rk}}{2} \left( \sigma_+ b_{rk} + \sigma_- b_{rk}^\dagger \right). \tag{1}$$

The first term is the qubit, $H_0 = (\Delta/2)\sigma_z$, where $\sigma_{x,y,z}$ are the Pauli operators and $\Delta$ is the qubit energy between the ground state $|g\rangle = |\sigma_z = -1\rangle$ and the excited state $|e\rangle = |\sigma_z = +1\rangle$. The second term denotes the heat baths modeled as a collection of harmonic oscillators.

The operator $b_{rk}$ ($b_{rk}^\dagger$) annihilates (creates) a boson of the heat bath $r$ in mode $k$ of energy $\omega_{rk}$. The heat bath $r$ is in thermal equilibrium characterized by the Bose-Einstein distribution $n_r(\omega) = (e^{\omega/T_r} - 1)^{-1}$ with the temperature $T_r$. The remaining term represents the coupling between the qubit and the heat bath $r$ with the interaction strength $\lambda_{rk}$. Here, $\sigma_\pm = (\sigma_x \pm i\sigma_y)/2$.

## 2.2 Measurement and feedback

We consider that the qubit pure state $|m\rangle$ is measured with the strength $\gamma$ and then transferred to a desired pure state $|n\rangle$ by a unitary operator

$$U_{nm} = |n\rangle\langle m| + |\bar{n}\rangle\langle\bar{m}|, \tag{2}$$

where $|\bar{m}\rangle$ and $|\bar{n}\rangle$ are the orthogonal states of $|m\rangle$ and $|n\rangle$, respectively, i.e., $\langle m|\bar{m}\rangle = \langle n|\bar{n}\rangle = 0$. For $|n\rangle = |m\rangle$, the unitary operator becomes the identity operator, $U_{nn} = I$, and then the quantum dynamics induced by the monitor is only the quantum measurement. The qubit pure state $|m\rangle$ is a state on the surface of the Bloch sphere and characterized by the polar angle $\theta_m$ and azimuthal angle $\phi_m$, as $|m\rangle = \cos(\theta_m/2)|g\rangle + e^{i\phi_m}\sin(\theta_m/2)|e\rangle$. When the measurement outcomes are discarded, the averaged dynamics of the density matrix $\varrho(t)$ under continuous measurement and feedback follow the Lindblad equation, $\dot{\varrho}(t) = -i[H, \varrho(t)] + \mathcal{D}_{\mathrm{M}}[\varrho(t)]$. The effects of the measurement and feedback are incorporated through [1, 3]

$$\mathcal{D}_{\mathrm{M}}[\varrho(t)] = \gamma\left[U_{nm}P_m\varrho(t)P_mU_{nm}^\dagger - \frac{1}{2}\{P_m, \varrho(t)\}\right], \tag{3}$$

where $P_m = |m\rangle\langle m|$ is the projection operator.

When the coupling between the qubit and the heat baths is weak, the Born-Markov approximation allows us to obtain the dynamics of the reduced density matrix $\rho(t) = \mathrm{tr}_{\mathrm{B}}[\varrho(t)]$, where $\mathrm{tr}_{\mathrm{B}}[\cdots]$ denotes the tracing out of the degrees of freedom of the heat baths, as

$$\dot{\rho}(t) = -i[H_0, \rho(t)] + \mathcal{D}_{\mathrm{B}}[\rho(t)] + \mathcal{D}_{\mathrm{M}}[\rho(t)], \tag{4}$$

where the dissipator to the heat baths is

$$\mathcal{D}_{\mathrm{B}}[\rho(t)] = \sum_r \mathcal{D}_{\mathrm{B},r}[\rho(t)] = \sum_{s=\pm}\Gamma_s\left(\sigma_{\bar{s}}\rho(t)\sigma_s - \frac{1}{2}\{\sigma_s\sigma_{\bar{s}}, \rho(t)\}\right), \tag{5}$$

with $\bar{s} = -s$. Here, $\Gamma_+ = \sum_r(\pi/2)I_r(\Delta)[1 + n_r(\Delta)]$ and $\Gamma_- = \sum_r(\pi/2)I_r(\Delta)n_r(\Delta)$ are the total emission and absorption rates of multiple heat baths, respectively, where $I_r(\omega) = \sum_k\lambda_{rk}^2\delta(\omega - \omega_{rk})$ is the spectral density for the heat bath $r$. In this work, we assume the Ohmic heat bath, $I_r(\omega) = 2\alpha_r\omega e^{-\omega/\omega_c}$ with the dimensionless coupling strength $\alpha_r$ and the cutoff frequency $\omega_c$ [43, 44]. Note that, to justify the Born-Markov approximation, the total emission and absorption rates $\Gamma_\pm$ must be much smaller compared with the qubit energy $\Delta$ [45]. As long as this condition is met, it does not matter if the heat baths have different temperatures.

## 3 Steady-state energy flow

We are interested in the energy flow from the monitor into the qubit [29, 42],

$$J(t) = \mathrm{tr}_0[H_0\mathcal{D}_{\mathrm{M}}[\rho(t)]] = \gamma\langle m|\rho(t)|m\rangle\,\mathrm{tr}_0[H_0P_n] - \frac{\gamma}{2}\mathrm{tr}_0[H_0\{P_m, \rho(t)\}], \tag{6}$$

where $\mathrm{tr}_0[\cdots]$ denotes the trace about the degrees of freedom of the qubit. The positive (negative) energy flow indicates the heating (cooling) of the qubit by the quantum measurement and feedback.

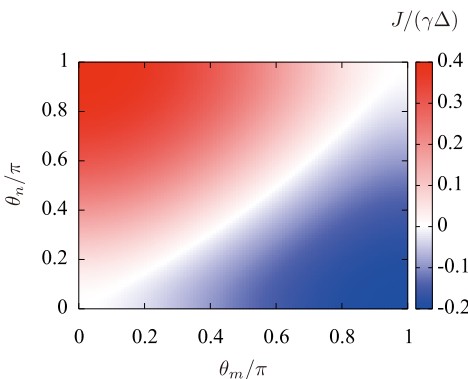

Figure 2: Steady-state energy flow as a function of the measurement state $\theta_m$ and the feedback state $\theta_n$ for $\Gamma_+/\Delta = 0.1$, $\Gamma_-/\Delta = 0.05$, and $\gamma/\Delta = 0.1$. These parameters correspond to $\alpha_{\text{eff}} \approx 0.0191$ and $T_{\text{eff}}/\Delta \approx 1.44$. For the measurement-only case $(\theta_m = \theta_n)$, the energy flow never takes negative values.

## 3.1 Minimum and maximum

At the steady-state limit, we can evaluate the steady-state energy flow by solving the Lindblad equation (4) for $\dot{\rho}(t) = 0$. For the weak coupling case ($\gamma$, $\Gamma_\pm \ll \Delta$), the steady-state energy flow is approximated as

$$J \approx \frac{\gamma\Delta}{2} \frac{\gamma_+(\cos\theta_n - \cos\theta_m) - \gamma_-(1 - \cos\theta_m \cos\theta_n)}{\gamma(\cos\theta_n \cos\theta_m - 1) - 2\gamma_+}, \tag{7}$$

where $\gamma_\pm = \Gamma_+ \pm \Gamma_-$, thus, $0 < \gamma_- < \gamma_+$. Note that the dependence of the steady-state energy flow (7) on $\phi_m$ and $\phi_n$ is a minor correction for the weak coupling case. Thus, the measurement and feedback states are characterized by $\theta_m$ and $\theta_n$, respectively. The steady-state energy flow satisfies the following inequality,

$$-\frac{\Gamma_-}{\gamma_+ + \gamma} \leq \frac{J}{\gamma\Delta} \leq \frac{\Gamma_+}{\gamma_+ + \gamma}. \tag{8}$$

The maximum and minimum energy flows $J_{\text{max/min}} = \pm\gamma\Delta\Gamma_\pm/(\gamma_+ + \gamma)$ occur at $(\theta_m, \theta_n) = (0, \pi)$ and $(\pi, 0)$, respectively. This inequality indicates that the qubit can be both heated and cooled by the monitoring and feedback.

For $(\theta_m, \theta_n) = (\pi, 0)$, the measurement is to the excited state and then feedback is to the ground state. In this process, the qubit energy decreases by $\Delta$, cooling the qubit. The steady-state energy flow takes minimum $J_{\text{min}} = -\gamma\rho_{ee}(\langle e|H_0|e\rangle - \langle g|H_0|g\rangle) = -\gamma\Delta\Gamma_-/(\gamma_+ + \gamma)$, where $\rho_{ij} = \langle i|\rho|j\rangle$. On the other side, for $(\theta_m, \theta_n) = (0, \pi)$, the measurement projects the system in the ground state and the feedback in the excited state. For this case, the qubit energy is increased by the measurement and the feedback, changing the qubit energy by $\Delta$. The steady-state energy flow is maximum and written as $J_{\text{max}} = -\gamma\rho_{gg}(\langle g|H_0|g\rangle - \langle e|H_0|e\rangle) = \gamma\Delta\Gamma_+/(\gamma_+ + \gamma)$.

Figure 2 shows the steady-state energy flow for $\Gamma_+/\Delta = 0.1$, $\Gamma_-/\Delta = 0.05$, and $\gamma/\Delta = 0.1$, which corresponds to the effective coupling strength $\alpha_{\text{eff}} \approx 0.0191$ and the effective temperature $T_{\text{eff}}/\Delta \approx 1.44$ when the effects of multiple heat baths are represented by an effective single heat bath. The steady-state energy flow reaches maximum at $(\theta_m, \theta_n) = (0, \pi)$ and minimum at $(\theta_m, \theta_n) = (\pi, 0)$, as expected, and its sign changes between them. The energy exchange does not occur at the edges $(\theta_m, \theta_n) = (0, 0)$ and $(\pi, \pi)$ and on the convex curve towards the measurement state $\theta_m$ connecting the edges. The convex curve indicates that the steady-state energy flow never takes negative ($J \geq 0$) for the measurement only case, $\theta_n = \theta_m$ [42].

Note that the energy flow can exceed the minimum and maximum values of the steady-state energy flow. When the qubit is prepared in the excited (ground) state for measurement to the excited (ground) state and feedback to the ground (excited) state, the energy flow reaches its minimum (maximum) value, $\mp\gamma\Delta$.

## 3.2 Symmetry at $\theta_m + \theta_n = \pi$

The steady-state energy flow is symmetric with $\theta_m + \theta_n = \pi$, corresponding to the feedback of the measurement state to the opposite side of the Bloch sphere, in the weak coupling case. On the axis of the mirror symmetry, $\theta_m + \theta_n = \pi$, the steady-state energy flow reads

$$J = \frac{\gamma\Delta}{2} \frac{\gamma_-(1+\cos^2\theta_m) + 2\gamma_+ \cos\theta_m}{\gamma(1+\cos^2\theta_m) + 2\gamma_+} . \tag{9}$$

This monotonically decreases from the maximum to the minimum energy flow as a function of $\theta_m$, and its sign changes at $\theta_m = \pi - \cos^{-1}[\tanh(\Delta/4T)]$ ($> \pi/2$). As the temperature decreases, the positive energy flow area is extended. This asymmetry between the positive and negative energy flows is controlled by the temperature of the heat baths coupled with the qubit. For $\theta_m = |\epsilon|$ and $\theta_m = \pi - |\epsilon|$, where the steady-state energy flow goes to maximum and minimum at $|\epsilon| \to 0$, respectively, the steady-state energy flow deviates quadratically from the maximum and minimum energy flows in the same way for $|\epsilon| \ll 1$,

$$\frac{J(\theta_m \approx 0)}{J_{\max}} \approx \frac{J(\theta_m \approx \pi)}{J_{\min}} \approx 1 - \frac{\Gamma_+ + \Gamma_-}{\Gamma_+ + \Gamma_- + \gamma} \frac{|\epsilon|^2}{2} . \tag{10}$$

## 3.3 Zero energy exchange

Finally, we consider the condition of the zero energy exchange ($J = 0$). At the edges, $(\theta_m, \theta_n) = (0, 0)$ and $(\pi, \pi)$, the measurement is commuting with the qubit Hamiltonian. The commuting measurement does not disturb the qubit state, and then the steady-state energy flow does not flow. When the feedback process is present, $|m\rangle \neq |n\rangle$, the qubit state is disturbed by the measurement and feedback process. Near the lower-left edge, where $\theta_m, \theta_n \ll 1$, the energy flow vanishes when $\theta_n = e^{-\Delta/(2T)}\theta_m$. Conversely, near the upper-right edge, where $\pi - \theta_m, \pi - \theta_n \ll 1$, energy flow ceases when $\pi - \theta_n = e^{+\Delta/(2T)}(\pi - \theta_m)$. As the temperature decreases, the curve in which the steady-state energy flow vanishes deviates more from the diagonal line ($\theta_n = \theta_m$), and then the positive energy flow area is extended.

Now, we consider the unraveling of the Lindblad equation (4), which describes the quantum dynamics of the reduced density matrix conditioned by the measurement outcomes, $\rho_c(t)$ [1],

$$\rho_c(t+\Delta t) = \rho_c(t) - i[H_0, \rho_c(t)]\Delta t - \mathcal{D}_B[\rho_c(t)]\Delta t + \mathcal{D}_M^{(1)}[\rho_c(t)]\Delta t + \mathcal{D}_M^{(2)}[\rho_c(t)]\Delta N_t , \tag{11}$$

where the measurement and feedback effects are decomposed into two contributions,

$$\mathcal{D}_M^{(1)}[\rho_c] = \gamma\langle m|\rho_c|m\rangle\rho_c - \frac{\gamma}{2}\{P_m, \rho_c\} , \tag{12a}$$

$$\mathcal{D}_M^{(2)}[\rho_c] = P_n - \rho_c . \tag{12b}$$

Here, $\Delta N_t$ is the Poisson increment and satisfies $\Delta N_t \Delta N_t = \Delta N_t$. It takes $\Delta N_t = 1$ when the monitor detects that the qubit is in $|m\rangle$ at time $t$ and $\Delta N_t = 0$ otherwise. The probability of $\Delta N_t = 1$ is given by $\mathbb{E}[\Delta N_t] = \gamma\langle m|\rho_c(t)|m\rangle\Delta t$, where $\mathbb{E}[\cdots]$ denotes the ensemble average over the measurement outcomes. The unraveled equation (11) reproduces the Lindblad equation (4) after the ensemble average, $\rho(t) = \mathbb{E}[\rho_c(t)]$. For $\Delta N_t = 1$, the conditional reduced

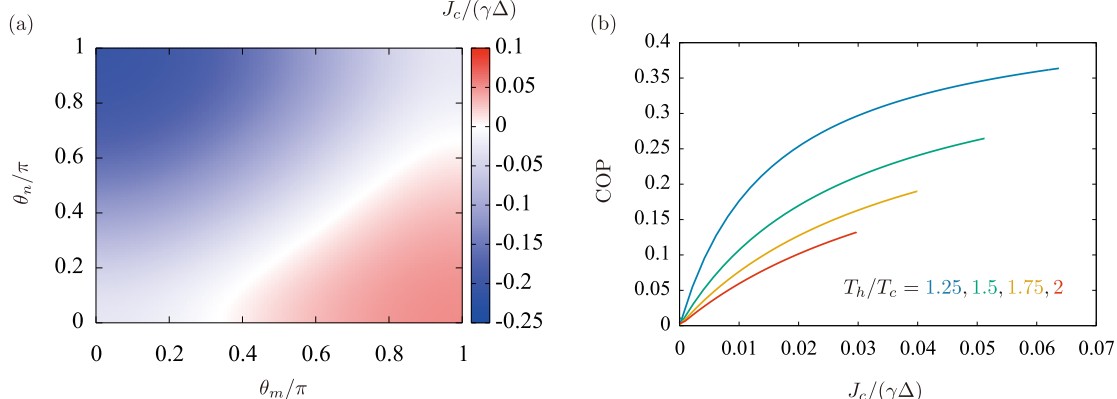

Figure 3: (a) Heat current from the cold heat bath to the qubit $J_c$ as a function of $\theta_m$ and $\theta_n$, obtained by solving the quantum master equation (4), for $T_h/\Delta = 1.5$, $T_c/\Delta = 1$, $\alpha_h = \alpha_c = 0.01$, and $\gamma/\Delta = 0.01$. The quantum measurement cooling occurs in the red region ($J_c > 0$). (b) The COP as a function of $J_c$ ($> 0$) at symmetric point $\theta_m + \theta_n = \pi$ for different temperature biases.

density matrix jumps to the feedback state, $\rho_c(t + \Delta t) = |n\rangle\langle n|$, called a quantum jump, and then the ensemble-averaged qubit energy at $t + \Delta t$ is $(\gamma\Delta/2)\rho_{mm}\text{tr}_0[H_0 P_n]\Delta t$. Since this energy change by the quantum jump is due to the quantum measurement and feedback, the energy flow induced by the monitor is $J^{(2)} = \gamma\rho_{mm}\text{tr}_0[H_0 P_n] - \gamma\rho_{mm}\text{tr}_0[H_0\rho]$. For $\Delta N_t = 0$, since the reduced density matrix evolves as $\dot{\rho} = -i[H_0, \rho] + \mathcal{D}_B[\rho] + \mathcal{D}_M^{(1)}[\rho]$, the energy flow by the monitor is $J^{(1)} = \text{tr}_0[H_0\mathcal{D}_M^{(1)}[\rho]] = \gamma\rho_{mm}\text{tr}_0[H_0 P_n] - (\gamma/2)\text{tr}_0[H_0\{P_m, \rho\}]$ [46]. Note that $J = J^{(1)} + J^{(2)}$. Therefore, the first term of the energy flow (6) represents the backaction of the detection ($\Delta N = 1$) and the second term the backaction of *no detection* ($\Delta N = 0$). The energy exchange does not occur between the qubit and the monitor when they are balanced, i.e., $J^{(1)} + J^{(2)} = 0$, corresponding to $\rho_{mm}(\cos\theta_m - \cos\theta_n) = \sin\theta_m\text{Re}[\rho_{m\bar{m}}]$. One can confirm that it reproduces $\theta_m = 0$ and $\pi$ for the measurement-only case ($\theta_m = \theta_n$).

## 3.4 Quantum measurement cooling

The continuous measurement and feedback can induce the qubit cooling ($J < 0$), as discussed in Sec. 3.1. This indicates the possibility of the quantum measurement cooling [22], which is the energy extraction from the colder heat bath by quantum measurement. Now, we consider the two heat baths ($r = h, c$) with different temperatures ($T_h < T_c$). The sign of the heat current from the cold heat bath determines if the quantum measurement cooling occurs. The heat currents from the cold and hot heat baths to the qubit are defined as

$$J_c(t) = \text{tr}_0[H_0\mathcal{D}_{B,c}[\rho(t)]], \tag{13a}$$

$$J_h(t) = \text{tr}_0[H_0\mathcal{D}_{B,h}[\rho(t)]], \tag{13b}$$

respectively. We demonstrate the steady-state heat current flowing out of the cold heat bath $J_c$ for different temperature biases in Fig. 3 (a). The region where the quantum measurement cooling occurs ($J_c > 0$) is observed, and smaller than the region of the qubit cooling ($J < 0$), as shown in Fig. 2.

The efficiency of the quantum measurement cooling is characterized by the coefficient of performance (COP) [5, 29, 47],

$$\text{COP} = \left|\frac{J_c}{J_h + J_c}\right|. \tag{14}$$

Figure 3 (b) shows the COP as a function of $J_c$ ($> 0$), i.e., the region where the quantum measurement cooling occurs, at $\theta_m + \theta_n = \pi$. The COP monotonically increases with increasing $J_c$ and decreasing the temperature bias.

## 4  Fluctuation

So far, we considered the steady-state energy flow after taking the ensemble average over the measurement outcomes. However, since the measurements are a stochastic process, the energy flow fluctuates around the averaged energy flow even at the steady-state limit. To characterize the fluctuation of the energy flow, we introduce the power spectrum as

$$S(\omega) = \frac{2}{\mathcal{T}} \int_{-\mathcal{T}/2}^{\mathcal{T}/2} dt \int_{-\mathcal{T}/2}^{\mathcal{T}/2} dt' \, e^{i\omega(t-t')} C_J(t, t'; t_0), \tag{15}$$

where $C_J(t, t'; t_0) = \mathbb{E}[\delta J_c(t + t_0)\delta J_c(t' + t_0)]$ is the correlation function of the energy flow fluctuation around the averaged energy flow, $\delta J_c(t) \equiv J_c(t) - J(t)$. The conditional energy flow $J_c(t)$ is decomposed into the contributions from the backaction of *no detection* ($\Delta N_t = 0$) and the quantum jump ($\Delta N_t = 1$),

$$J_c^{(1)}(t)\Delta t = \mathrm{tr}_0\left[ H_0 \mathcal{D}_{\mathrm{M}}^{(1)}[\rho_c(t)] \right]\Delta t, \tag{16a}$$

$$J_c^{(2)}(t)\Delta t = \mathrm{tr}_0\left[ H_0 \mathcal{D}_{\mathrm{M}}^{(2)}[\rho_c(t)] \right]\Delta N_t, \tag{16b}$$

respectively. Here, $t_0$ is large enough that the qubit reaches the steady state and the time $\mathcal{T}$ is sufficiently long for the current correlation to vanish and will eventually be taken to infinity. Below, we address the correlation function of the energy change during the infinitesimal time $\Delta t$,

$$C(t, t'; t_0) = \mathbb{E}[\delta Q_c(t + t_0)\delta Q_c(t' + t_0)] = C_J(t, t'; t_0)(\Delta t)^2, \tag{17}$$

where the conditional energy change is $Q_c(t) = Q_c^{(1)}(t) + Q_c^{(2)}(t) = J_c^{(1)}(t)\Delta t + J_c^{(2)}(t)\Delta t$.

Now, the qubit is in the steady state at $t_0$, and thus the correlation function depends only on the time difference $t - t'$. Then, the power spectrum is rewritten as

$$S(\omega) = \frac{2}{(\Delta t)^2} \int_{-\mathcal{T}}^{\mathcal{T}} dt \, e^{i\omega t} C(t). \tag{18}$$

We note that we finally take $\Delta t \to 0$ after performing the integration. The correlation function is decomposed in a delta-function term, coming from the time local correlation, and a time non-local correlation function,

$$C(t) = C_0 \Delta t \, \delta(t) + C_1(t \neq 0). \tag{19}$$

Since the non-local term is the even function, $C_1(t) = C_1(-t)$, the power spectrum consists of the frequency-independent and the frequency-dependent terms as $S(\omega) = S_0 + S_1(\omega)$,

$$S_0 = \frac{2}{\Delta t} C_0, \tag{20a}$$

$$S_1(\omega) = \frac{4}{(\Delta t)^2} \int_0^\infty dt \, \cos(\omega t) C_1(t). \tag{20b}$$

We note that we here took $\mathcal{T} \to \infty$ because $C_1(t)$ vanishes for a sufficiently long time. In this work, we focus on the condition $\tau_0 \gg \tau_r$, where $\tau_0 = (\gamma \rho_{mm})^{-1}$ is the average time interval between the measurements and $\tau_r$ is the relaxation time it takes for the qubit to reach the steady state from a pure state due to dissipation with heat baths. This region means that the measurements are rare events compared with the relaxation time scale, and it is justified for $\gamma \ll \Gamma_\pm$ [46].

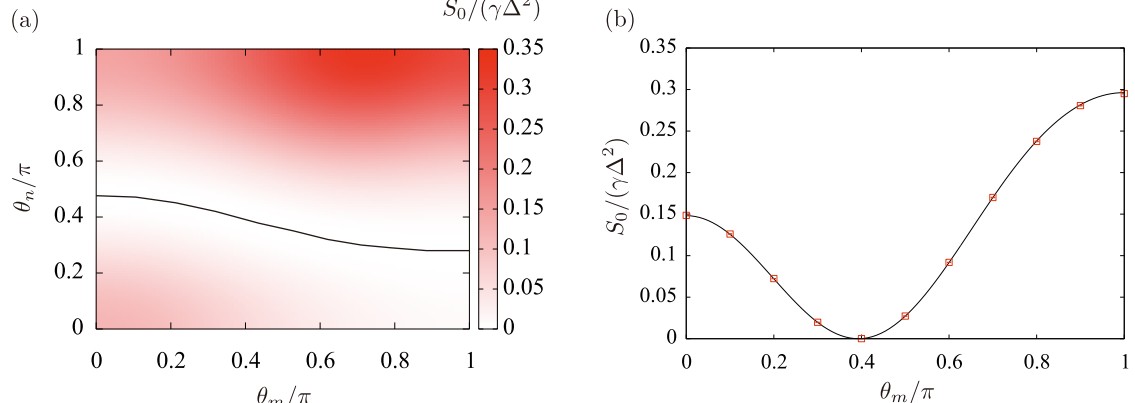

Figure 4: (a) Poisson noise $S_0$ as a function of the measurement state $\theta_m$ and the feedback state $\theta_n$ for $\Gamma_+/\Delta = 0.3$, $\Gamma_-/\Delta = 0.15$, and $\gamma/\Delta = 0.01$, obtained by Eq. (21) and the numerical simulation for $\langle\sigma_z\rangle$ using the quantum master equation (4). The solid line represents $S_0 = 0$. (b) $\theta_m$ dependence of the Poisson noise for the measurement-only protocol, which corresponds to the diagonal line ($\theta_m = \theta_n$) in panel (a). The solid line represents the analytical formula (22) and the plots are obtained by the numerical simulation using the stochastic master equation (11) with $3.2 \times 10^6$ trajectories.

## 4.1 Poisson noise

The frequency-independent term (20a) of the power spectrum comes from the local correlation in time, $C(0) = \mathbb{E}\left[Q_c^2\right] - \mathbb{E}[Q_c]^2$. In the following, we drop the time index for the steady state. At $\Delta t \to 0$, the correlation of the energy changes by the quantum jump process, $\mathbb{E}[(Q_c^{(2)})^2]$ is predominant. Therefore, the frequency-independent power spectrum is obtained as

$$S_0 \approx 2\gamma\rho_{mm}Q_{\text{jump}}^2 = 2Q_{\text{jump}}J^{(2)}, \tag{21}$$

where $Q_{\text{jump}} = \text{tr}_0[H_0(P_n - \rho)]$ and $J^{(2)} = \mathbb{E}[Q_c^{(2)}]/\Delta t = \gamma\rho_{mm}\text{tr}_0[H_0(P_n - \rho)]$ are the energy change and the steady-state energy flow induced by the quantum jump process. This expression corresponds to the Poisson noise for non-interacting electrons in a one-dimensional nanowire with weak transmission; $S_{\text{Poisson}}(t) \approx 2eI(t)$, where $I(t)$ represents the mean electronic current [31, 32]. In this work, we call the frequency-independent contribution of the power spectrum $S_0$ as the "Poisson" noise. In the weak coupling regime, the Poisson noise is approximated as

$$S_0 \approx \frac{\gamma\Delta^2}{4}(\langle\sigma_z\rangle + \cos\theta_n)^2(1 - \langle\sigma_z\rangle\cos\theta_m), \tag{22}$$

with $\langle\sigma_z\rangle \approx -[2\gamma_- - \gamma(\cos\theta_m - \cos\theta_n)]/[2\gamma_+ + \gamma(1 - \cos\theta_m\cos\theta_n)]$.

Figure 4 (a) shows the Poisson noise for $\Gamma_+/\Delta = 0.3$, $\Gamma_-/\Delta = 0.15$, and $\gamma/\Delta = 0.01$. The Poisson noise varies, depending on the measurement and feedback states, and vanishes, as shown in the solid line, when the qubit energy of the steady state matches that of the feedback state, $\text{tr}_0[H_0\rho] = \text{tr}_0[H_0P_n]$. The line of $S_0 = 0$ is almost independent of the measurement state. The energy change due to the quantum jump is expressed as $Q_{\text{jump}} = -(\Delta/2)(\cos\theta_n + \langle\sigma_z\rangle)$, and the $\gamma$ dependence of $\langle\sigma_z\rangle$ is small for $\gamma \ll \gamma_\pm$, i.e., it is primarily determined by the dissipation due to the heat baths. Thus, $Q_{\text{jump}}$ is almost independent of the measurement state, while it strongly depends on the feedback state. Since $\langle\sigma_z\rangle < 0$ and $\langle\sigma_z\rangle$ is almost independent of $\gamma$, equivalently $\theta_m$ and $\theta_n$, $|Q_{\text{jump}}|$ takes maximum

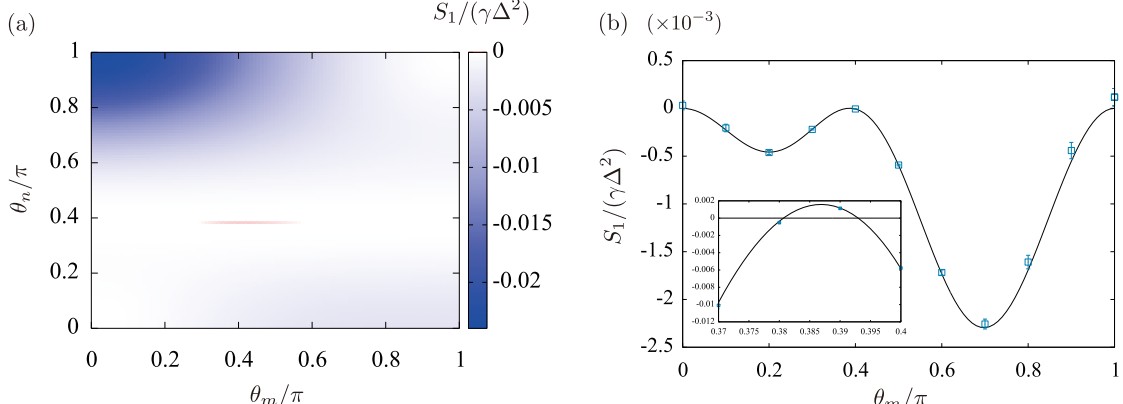

Figure 5: (a) The backaction noise $S_1$ as a function of $\theta_m$ and $\theta_n$ using Eq. (27) and the numerical simulation of the quantum master equation (4). The parameters are the same as Fig. 4. (b) For the measurement-only case ($\theta_m = \theta_n$), the measurement state $\theta_m$ dependence of the backaction noise. The solid line is obtained in the same way as the panel (a) and plots are obtained by the numerical simulation using the stochastic master equation (11) with $3.7 \times 10^8$ trajectories. The inset shows an enlarged view near $S_1 = 0$.

at $\theta_n = \pi$ and its value is positive, $Q_{\text{jump}} > 0$. Concerning another contribution of the Poisson noise, $\rho_{mm}$, because $\rho_{mm} \approx (1 - \langle \sigma_z \rangle \cos \theta_m)/2$ for $\gamma \ll \gamma_\pm$, it is almost independent of the feedback state and reaches maximum at $\theta_m = 0$. Therefore, the Poisson noise has a maximum value at $\theta_m = 0$ and $\theta_n = \pi$, as shown in Fig. 4. There, more energy is transferred between the qubit and the monitor more frequently.

We also show the $\theta_m$ dependence of the Poisson noise for the measurement-only protocol, $|m\rangle = |n\rangle$, in Fig. 4 (b). The expression for the Poisson noise (22), describes the numerical simulation well. The Poisson noise is not symmetric with respect to $\theta_m = \pi/2$, unlike the steady-state energy flow. This is reflected in the characteristic statistics induced by the continuous measurement and feedback, which differs from the case of the electronic current. The resulting fluctuations provide direct access to the quantum jump, which is indistinguishable from the backaction of no detection at the level of the energy exchange. The characteristic statistics arise from the fact the probability of a measurement occurring depends on the qubit state, and the energy exchange is also affected by the measurement backaction, discussed in Sec. 4.2.

## 4.2 Backaction noise

After the quantum jump, energy exchanges between the qubit and the monitor to reach the steady state as a measurement backaction. The energy flow fluctuation induced by the measurement backaction is characterized by the non-local correlation in time, $C_1(t)$, providing the frequency-dependent contribution of the power spectrum $S_1(\omega)$, called the *backaction noise*. At the dc limit ($\omega = 0$), the power spectrum is written as

$$S_1 \equiv S_1(0) = \frac{4}{(\Delta t)^2} \int_0^\infty dt\, \mathbb{E}[\delta Q_c(t+t_0)\delta Q_c(t_0)]. \tag{23}$$

Under the condition, $\tau_0 \gg \tau_r$, we assume the quantum jumps are independent of each other and the qubit just before the quantum jump is in a steady state. The dynamics there can be interpreted as the relaxation from the post-selected state, $P_n$. When the quantum jump

occurs at $t_0$, the backaction noise is approximated as

$$S_1(\Delta N_{t_0} = 1) \approx \frac{4}{\Delta t} Q_{\text{jump}} Q_{\text{ex}}, \tag{24}$$

where we took $\Delta t \to 0$ limit. Here, $Q_{\text{ex}} = \int_0^\infty dt [J(t; 0) - J]$ is the excess energy from the post-selected state, i.e., $\rho(0) = P_n$, and is a quantity related to the transient dynamics. On the other hand, when the quantum jump does not occur at $t_0$ and occurs at $t_{-1}$ ($< t_0$), the backaction noise is approximately given by

$$S_1(\Delta N_{t_0} = 0) \approx 4[J(t_0; t_{-1}) - J] \int_0^\infty dt \, [J(t + t_0; t_{-1}) - J]. \tag{25}$$

Hence, the approximated form of the backaction noise is obtained as

$$S_1 \approx P(\Delta N_{t_0} = 1) S_1(\Delta N_{t_0} = 1) + \sum_{t_{-1}=0}^{t_0 - \Delta t} P(\Delta N_{t_{-1}} = 1) S_1(\Delta N_{t_0} = 0) \tag{26}$$

$$\approx 4\gamma \rho_{mm} \left( Q_{\text{jump}} + \frac{1}{2} Q_{\text{ex}} \right) Q_{\text{ex}}, \tag{27}$$

at $\Delta t \to 0$, where $P(\Delta N_{t_0} = 1) = 1 - P(\Delta N_{t_0} = 0) = \gamma \rho_{mm}(t_0) \Delta t$ is the probability of the quantum jump at $t_0$.

We plot the backaction noise in the plane of $(\theta_m, \theta_n)$ in Fig. 5 (a) for the same parameters of the Poisson noise in Fig. 4. The magnitude of the backaction noise is smaller than the Poisson noise by one order, as the excess energy is on the order of $\Delta\gamma/\gamma_+$ and $Q_{\text{jump}}$ is on the order of $\Delta$. The backaction noise is negative in a large range of $(\theta_m, \theta_n)$. The negative backaction noise indicates that, from Eq. (27), the energy change by the quantum jump and the excess energy have opposite signs. This is the measurement backaction; when the qubit energy is stored from the monitor by the detection ($Q_{\text{jump}} > 0$), less energy flows out of the monitor to the qubit in the dynamics after the detection compared with the steady-state energy flow ($Q_{\text{ex}} < 0$), and vice versa.

The backaction noise has a minimum value around $\theta_m = 0$ and $\theta_n = \pi$. The excess energy is the difference between the total energy transferred from the monitor to the qubit until the post-selected state reaches the steady state and the steady-state energy flow during that time. Roughly speaking, $|Q_{\text{ex}}|$ becomes larger as $J(t = 0; 0) = -(\gamma\Delta/4)\sin(\theta_m - \theta_n)\sin\theta_n$ and $J$ are more different because, for $\gamma \ll \gamma_\pm$, the relaxation rate to the steady-state energy flow is dominantly determined by the heat baths, and the measurement effect to the rate is small. The oscillation of the transient energy flow due to the quantum coherence affects the excess energy, but it reduces $|Q_{\text{ex}}|$ because the integral of the product of a trigonometric function and an exponential decay function becomes smaller than that without oscillation when the oscillation frequency, $\sim \Delta$ is larger than the relaxation rate, $\sim \gamma_+$. At $\theta_m = 0$ and $\theta_n = \pi$, $J$ ($> 0$) takes maximum as discussed in Sec. 3 and the energy flow relaxes to the steady-state one without the oscillation while $J(t = 0; 0) = 0$. Then, $|Q_{\text{ex}}|$ is maximum and $Q_{\text{ex}}$ is negative around $\theta_m = 0$ and $\theta_n = \pi$. Hence, from the fact that $\rho_{mm} Q_{\text{jump}}$ is positive and its absolute value has a maximum value at $\theta_m = 0$ and $\theta_n = \pi$, as discussed in Sec. 4.1, the backaction noise is negative and its absolute value reaches maximum around $\theta_m = 0$ and $\theta_n = \pi$. In contrast, when $\theta_m = \theta_n = 0$ or $\pi$, the excess energy vanishes, resulting in $S_1 = 0$, because the measurement commutes with the qubit Hamiltonian. In addition, the backaction noise takes 0 around $\theta_n/\pi \approx 0.4$, corresponding to $Q_{\text{ex}} = 0$ and $Q_{\text{jump}} + Q_{\text{ex}}/2 = 0$, which will be discussed below.

As mentioned above, the backaction noise has negative values for a large range of $(\theta_m, \theta_n)$, but it shows the small positive value, the order of $10^{-6}$, at $\theta_m/\pi \approx 0.3 - 0.5$ and $\theta_n/\pi \approx 0.4$

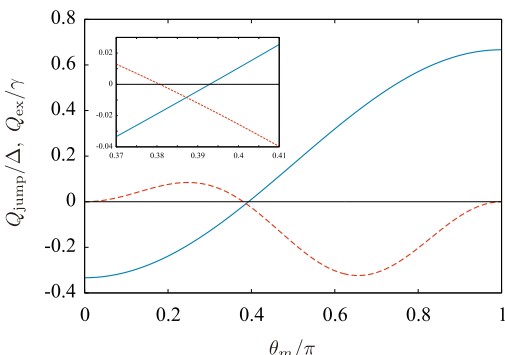

Figure 6: Energy change due to the quantum jump, $Q_{\text{jump}}$ (solid line), and excess energy, $Q_{\text{ex}}$ (dashed line), as a function of $\theta_m$ for the measurement-only case ($\theta_m = \theta_n$). The parameters are the same as Fig. 4. For a large range of $\theta_m$, $Q_{\text{jump}}$ and $Q_{\text{ex}}$ have opposite signs, but they are both negative when they are crossing 0. The inset shows an enlarged view where both $Q_{\text{jump}}$ and $Q_{\text{ex}}$ are negative near $\theta_m/\pi \approx 0.39$. The difference between $Q_{\text{jump}}$ and $Q_{\text{jump}} + Q_{\text{ex}}/2$ is not visible in this plot.

near $S_0 = 0$. It, here, violates $Q_{\text{jump}}Q_{\text{ex}} < 0$. This is attributed that the energy loss or gain of the qubit due to quantum measurement can also be compensated by the heat baths during the transient dynamics. Therefore, when focusing on the energy exchange between the monitor and the qubit, it is possible to slightly violate $Q_{\text{jump}}Q_{\text{ex}} < 0$ around $Q_{\text{jump}}, Q_{\text{ex}} \approx 0$. We plot $Q_{\text{jump}}$ and $Q_{\text{ex}}$ for the measurement-only case in Fig. 6, and it is indeed observed that both values are negative within the small range of $\theta_m/\pi \approx 0.381 - 0.393$.

Figure 5 (b) shows the backaction noise for the measurement-only case. The numerical simulation using the stochastic master equation (11) is in good agreement with the analytical formula (27). It crosses zero at four points. At the edges, $\theta_m = 0$ and $\pi$, the measurements commute with the qubit Hamiltonian, resulting in the $Q_{\text{ex}} = 0$, as discussed above. The remaining two points, $\theta_m/\pi \approx 0.381$ and $0.393$, correspond to $Q_{\text{ex}} = 0$ and $Q_{\text{jump}} = 0$, respectively (see Fig. 6).

## 4.3 Fano factor

Finally, let us compare the Poisson noise $S_0$ with the backaction noise $S_1$. To this end, we introduce the "Fano" factor, $\mathcal{F} = S(0)/S_0 = (S_0 + S_1)/S_0$, from the analogy of the electronic current [31, 32]. Using Eqs. (21) and (27), we obtained the Fano factor as

$$\mathcal{F} \approx \left( 1 + \frac{Q_{\text{ex}}}{Q_{\text{jump}}} \right)^2 . \tag{28}$$

For a large range of $(\theta_m, \theta_n)$, it is less than 1, i.e., sub-Poisson fluctuation, because $S_1 < 0$, where the deviation from the unity is on the order of $1 - \mathcal{F} \sim \mathcal{O}(\gamma/\gamma_+)$. The sub-Poisson statistics arise from the fact that the probability of a quantum jump depends on the qubit state, making the quantum jumps not entirely random in time. This phenomenon is analogous to electronic current, where electrons do not flow completely randomly due to the Pauli exclusion principle and Coulomb interactions. When $Q_{\text{jump}} = 0$, the Fano factor diverges because $Q_{\text{jump}}$ and $Q_{\text{ex}}$ do not become zero simultaneously, as discussed in Sec. 4.2. In the vicinity of $Q_{\text{jump}} = 0$, since $Q_{\text{jump}}$ and $Q_{\text{ex}}$ can have the same sign over a small range of $(\theta_m, \theta_n)$, the Fano factor is strongly enhanced, exceeding unity, i.e., super-Poisson fluctuation. At $\theta_m = \theta_n = 0$ and $\pi$, the fluctuation becomes the Poisson, $\mathcal{F} = 1$, because the commuting measurement does not induce the backaction noise $S_1 = 0$.

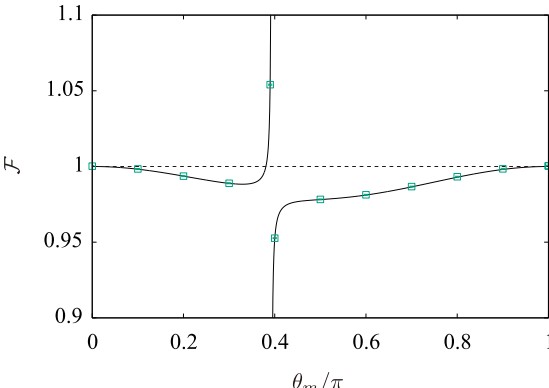

Figure 7: Fano factor as a function of the measurement state $\theta_m$ for the measurement-only case. The parameters are the same as Figs. 4 and 5. The solid line represents the analytical formula (28) and plots are obtained by the numerical simulation using the stochastic master equation (11) with $3.7 \times 10^8$ trajectories. The dashed line represents $\mathcal{F} = 1$.

Figure 7 shows the numerical simulation for the Fano factor using the same parameters of the Poisson and backaction noises shown in Figs. 4 and 5 for the measurement-only case. The analytical formula (28) is in good agreement with the numerical simulation, and it shows the sub-Poisson fluctuation for a large range of $\theta_m$. In the vicinity of $Q_{\text{jump}} = 0$ at $\theta_0/\pi \approx 0.395$, the Fano factor is strongly suppressed/enhanced as $\theta_m \to \theta_0 \pm 0$ because $Q_{\text{jump}}$ and $Q_{\text{ex}}$ have the same signs for $\tilde{\theta}_0 < \theta_m < \theta_0$, where $Q_{\text{ex}} = 0$ at $\tilde{\theta}_0/\pi \approx 0.381$, as discussed in Sec. 4.2.

## 5 Summary

We study the statistical property of energy exchange between the monitor and the dissipative qubit under continuous measurement and feedback. The first half is the steady-state energy flow when changing the measurement and feedback states. The energy flow can take a wider range of values compared with the measurement-only case. When doing the measurement to the ground state and the feedback to the excited state, energy maximally flows from the monitor to the qubit, which exceeds the maximum value for the measurement-only case. We also observe the qubit cooling by the measurement and the feedback, which is never seen for the measurement-only case. We identify the boundary between the cooling and the heating is determined by the temperature, the cooling region is extended as the temperature decreases.

The second part of this paper is devoted to the fluctuation of the energy flow by unraveling the Lindblad equation, the stochastic master equation. The power spectrum is composed of the frequency-independent and -dependent contributions, coming from the quantum jump and the measurement backaction dynamics, called the Poisson noise and the backaction noise, respectively, in this work. Both noises are characteristic of the continuous measurement process, different from the standard shot noise such as in the electronic circuit. For a large range of parameters, these spectra have the opposite sign, which can be interpreted as the measurement backaction and leads to the sub-Poisson Fano factor. However, around zero energy change by the quantum jump, these spectra have the same sign due to the dissipation to the heat baths, resulting in the super-Poisson Fano factor.

Our findings are helpful for the development of future applications in measurement-based thermal machines, such as qubit cooling and quantum refrigerators, and the further understanding of quantum thermodynamics in quantum dissipative systems from the aspect of

the current fluctuations. Recently, thanks to the advancements in fabrication techniques for nanoscale devices and improvements in heat measurement technologies [48,49], it is expected that the flow of quantum heat and its fluctuations can be directly observed by monitoring the temperature of the thermal bath.

The numerical results for the trajectory simulations used in Figs. 4 (b), 5 (b), and 7 are published on Zenodo [50].

# Acknowledgments

The authors thank the Supercomputer Center, the Institute for Solid State Physics, the University of Tokyo for the use of the facilities.

**Funding information** This work was supported by the JST Moonshot R&D- MILLENNIA Program Grant No. JPMJMS2061 and Y.T. acknowledges support by JSPS KAKENHI Grant No. 23K03273.

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
