# Peer review of "Energy exchange and fluctuations between a dissipative qubit and a monitor under continuous measurement and feedback"

_SciPost Physics, doi:SciPost Phys. Core 8, 016 (2025)_

## Round 1 · Referee Report · Anonymous (Referee 2) · 2025-1-9

Strengths

1- Clearly written 2- Interesting topic

Report

The authors have taken into account the criticism presented by the three Referees to improve the manuscript. I can recommend the present version for publication.

Recommendation

Publish (meets expectations and criteria for this Journal)

---

## Round 1 · Referee Report · Anonymous (Referee 1) · 2025-1-11

Strengths

  • Targets a timely topic in quantum thermodynamics.

Weaknesses

  • A novel link between different research areas is not elaborated convincingly enough.

Report

I thank the authors for the effort they put in improving the manuscript, following the suggestions by all referees. In particular, referring to my previous comments: - the statements about electronic noise are now more precise; - the addition of the COP analysis is useful and provides a starting point to compare the performance of the proposed setup with further quantifiers, such as the thermodynamic uncertainty relation; - I greatly appreciate the publication of the data on the quantum trajectory analysis.

Despite these improvements, I stand by the assessment in my previous report that the acceptance criterium mentioned by the authors is not quite met. As I previously mentioned, I do not think that the analysis of the noise and Fano factor qualifies as a novel and synergetic link between different research areas. Indeed, the authors consider a quantity (the current noise) which is also standardly investigated in electronic transport, explaining why in their setup it can show both sub- and super-Poissonian features. However, it is not clear to me how the insights from their analysis are beneficial for electronic transport settings and viceversa. I would also like to point out that the importance of analysing the fluctuations is well recognised in several works in quantum thermodynamics. While I acknowledge that this analysis is novel for the specific model investigated by the authors, I still fail to see that the authors' work provides the claimed novel and synergetic link.

In conclusion, I think this manuscript is a valuable contribution to the quantum thermodynamics community, but I believe it is more appropriate to recommend publication in SciPost Physics Core, whose acceptance criteria are met.

Recommendation

Accept in alternative Journal (see Report)

---

## Round 1 · Author Response

We thank all the referees for carefully reading our manuscript and providing useful comments. We have addressed all the comments point by point and revised our manuscript accordingly for resubmission, based on the referees’ comments. Please check our detailed replies in the “Reports on this Submission” section on our “SciPost Submission Page”.
We have attached the revised version of our manuscript in PDF format to this correspondence. The revisions in the resubmitted manuscript are highlighted in blue for clarity.

---

## Round 1 · List of Changes

• First paragraph on page 2: We have improved the sentence about the dissipation effects in continuous measurement.
  • Second paragraph on page 2: We have revised the explanation of the Poisson noise for the electronic current.
  • Eq. (5) and below on page 4: We have added a comment to clarify that our setup is valid for multiple heat baths.
  • Below Eq. (7) on page 5: we have included an explanation of θ_m and θ_n.
  • End of page 5: We have modified the sentence about the effective coupling strength and the effective temperature.
  • Caption of Fig. 2 on page 5: We have added values for the effective coupling strength and the effective temperature.
  • Beginning of page 6: We have added a discussion about the bounds of the heat current in the transient region.
  • Page 7: We have introduced a new Section 3.4 about quantum measurement cooling, along with new figures as Fig. 3.
  • Eq. (16) on page 8: We have added the definition of the conditional heat current.
  • Below Eq. (21) on page 10: We have revised the explanation of the Poisson noise for the electronic current.
  • End of Section 4.1 on page 10: We have improved the discussion on the comparison with the electronic current.
  • Below Eq. (28) on page 12: We have added a discussion about sub-Poisson statistics.
  • End of Section 5 on page 14: We have included a comment about the possibility of experimental detection.
  • End of Section 5 on page 14: We have provided information on Zenodo, where we have uploaded our numerical data.
  • We have replaced “heat current” with “energy exchange” or “energy flow”.
  • We have avoided using the term “bound” to prevent ambiguity.

---

## Editorial Decision

published